# Challenging an old paradigm by demonstrating transition metal-like chemistry at a neutral nonmetal center

David Biskup [1], Gregor Schnakenburg [1], René T. Boeré [2], Arturo Espinosa Ferao [3] ✉ & Rainer K. Streubel [1] ✉

We describe nonmetal adducts of the phosphorus center of terminal phosphinidene complexes using classical C- and N-ligands from metal coordination chemistry. The nature of the L-P bond has been analyzed by various theoretical methods including a refined method on the variation of the Laplacian of electron density $\nabla^2\rho$ along the L-P bond path. Studies on thermal stability reveal stark differences between N-ligands such as *N*-methyl imidazole and C-ligands such as *tert*-butyl isocyanide, including ligand exchange reactions and a surprising formation of white phosphorus. A milestone is the transformation of a nonmetal-bound isocyanide into phosphaguanidine or an acyclic bisaminocarbene bound to phosphorus; the latter is analogous to the chemistry of transition metal-bound isocyanides, and the former reveals the differences. This example has been studied via cutting-edge DFT calculations leading to two pathways differently favored depending on variations in steric demand. This study reveals the emergence of organometallic from coordination chemistry of a neutral nonmetal center.

In the 19th century, Werner reported his ground-breaking coordination theory of metal atoms and ions[1], one of the most important concepts in modern chemistry. Initially, a large range of complexes $ML_n$ where the metal is coordinated by neutral and anionic ligands L such as water, ammonia and halides were explored (Fig. 1), while elaborate organic N-ligands such as amines were added later, followed by P-ligands such as phosphanes and other even more elaborate ligand structures[2]. The important emergence of organometallic chemistry extended C-ligands L to carbon monoxide, isocyanides, carbenes and even arenes with a direct carbon-metal bond, all of which have enriched numerous chemical opportunities including homogeneous catalysis and many other applications[2]. Transition metal complexes with readily available coordination sites become prone to oxidative addition reactions which is a vital capability, especially, in catalysis[3–7]. Transition metal complexes[8–10], in general, display a wide variety of reactivity, which may be illustrated by isocyanide complexes: many

reactions are metal-centered such as substitution[11,12], redox[13–16], or M-L insertion[17], while others are ligand-centered such as nucleophilic addition[18–20]. An important early example of ligand transformation was the addition of amines to metal-bound isocyanides at the C-N bond to form carbene metal complexes[21–23]. The latter may be regarded not only as a fundamental change of the nature of the M-C bond, but also as the emergence of the field of organometallic chemistry from coordination chemistry.

A similar reactivity is highly uncharacteristic of main group metalloid and nonmetal compounds, save for some recent advances which will be briefly reviewed, focusing on group 13 and 14 metalloids and thereafter for nonmetals from group 15. Some rare but well investigated examples of neutral, low-coordinate metalloids include nitrile adducts of silylenes, including (transient) *end*-on ($\eta^1$) or *side*-on ($\eta^2$) coordinated nitriles reported by Tilley for a cationic silylene ruthenium(II) complex[24] and by Tokitoh for a neutral free silylene[25].

[1]Institut für Anorganische Chemie, Rheinische Friedrich-Wilhelms-Universität Bonn, Gerhard-Domagk-Str. 1, 53121 Bonn, Germany. [2]Department of Chemistry and Biochemistry, University of Lethbridge, Lethbridge, AB T1K3M4, Canada. [3]Departamento de Química Orgánica, Facultad de Química, Campus de Espinardo, Universidad de Murcia, 30100 Murcia, Spain. ✉e-mail: artuesp@um.es; r.streubel@uni-bonn.de

**a** $ML_n \rightleftharpoons ML_{n-1} + L$

**b** $ML_n + L' \rightleftharpoons ML_{n-1}L' + L$

**c** $ML_n + Nu \rightleftharpoons L_{n-1}M-L^{\ominus}$ ⟷ $L_{n-1}M=L^{\ominus}$
with $Nu^{\oplus}$ substituents.

**Fig. 1 | Typical reactivity of transition metal complexes. a** Dissociation, **b** ligand substitution reactions and **c** reactions at the ligand of transition metal complexes (Nu: nucleophile).

The same author also reported an example of a stable *end*-on isocyanide-to-silylene adduct using sterically demanding substituents at N[26,27]. This followed an early hypothesis by Weidenbruch, who had proposed such short-lived adducts as intermediates years before[28,29]. Tokitoh later demonstrated that the isocyanide ligand can be replaced by a different isocyanide, thus proving an equilibrium between the R₂Si·CNR adduct and the free silylene and isocyanide[30]. Iwamoto and Kira also synthesized dialkylsilaketenimines without aromatic substituents at silicon, which possess a slightly bent cumulene structure in the solid state and a weak Si-C bond, i.e., dissociation occurs in solution above −30 °C[31]. Very recently, this field was expanded by Filippou who demonstrated that an isocyanide can be replaced in an oxidative fashion by reaction of (SIDipp)-Si(CNAr^Mes) (SIDipp = 1,3-bis(2,6-diisopropylphenyl)imidazolidin-2-ylidene, Ar^Mes = 2,6-bis(2,4,6-trimethylphenyl)phenyl) with Ge(Ar^Mes)Cl[32]. To the best of our knowledge, isosteric silylene adducts of carbon monoxide, R₂Si·CO, have only been investigated by computation and/or detected in cryogenic matrices[33–36]. In 2002, Tokitoh reported the formation of azasilepines from the reaction of a silylene with pyridine or 4-(dimethylamino)pyridine (DMAP), with proposed Si-adducts as intermediates[37]. Recently, Inoue described the synthesis and isolation of stable DMAP-to-silylene adducts. Under elevated temperatures, these adducts undergo 1,1 additions of dihydrogen and 1,2 additions of ethene. In the absence of trapping reagents, the DMAP adducts decompose to disiletanes, azasilepines or cyclotrisilanes, depending on the substituents at Si[38].

Adducts of low-coordinate boron centers representing group 13 metalloids seem to be rare[39–43]. But classical boranes, having all valence electrons involved in covalent bonding, can also form Lewis acid adducts with nitriles, isocyanides and carbon monoxide due to the vacant *p*-orbital. Of special interest is one report from B,C-cluster chemistry, in which a nitrile-to-carborane adduct reacts with secondary amines by nucleophilic addition to form acetamidine derivatives[40].

To the best of our knowledge, the chemistry of nonmetal main group element adducts at a low-coordinate phosphorus center started with a report by Burford on a (cationic) phosphadiazonium σ¹λ³-P-center, which was later expanded by the same author with studies on adduct formation at a (cationic) σ²λ²-phosphenium, including ligand substitution reactions[44–55]. Strained cyclic compounds such as (cationic) phosphirenium salts, can also undergo exchange reactions of the alkyne moiety, as demonstrated by Wild[56,57].

Our group has described evidence for the formation of transient *end*-on adducts of nitriles, carbonyls, thioketones, thiourea, imines, *N*-heterocyclic carbenes (NHCs), and some isolated examples of isocyanides and carbodiimides, to neutral electrophilic terminal phosphinidene complexes bearing a bulky P-substituent[58–67]. Later, Mathey proposed that *N*-methylimidazole (*N*-MeIm) can bind transiently *end*-on to electrophilic terminal phosphinidene molybdenum complexes[68,69] which, finally, produced cyclo(oligo)phosphanes and [Mo(CO)₃(*N*-MeIm)₃] in the absence of trapping reagents[70]. Various families of *end*-on P-adducts of ligands (C-, N-, O- and halide donors) bound to phosphorus of terminal electrophilic phosphinidene complexes have recently been studied computationally[71–73]. The latest addition to this emerging field came from the group of Bertrand, who

demonstrated that a sterically encumbered phosphanyl phosphinidene R₂PP, accessed via a multistep synthetic protocol, can form a *P*-PR₃ adduct. The latter can undergo replacement by stronger ligands, such as carbenes and isocyanides, resulting in mono-phosphacarbodiimides or phosphaalkenes, respectively[74].

Despite these singular recent advances, the scope of and knowledge about the chemistry of such nonmetal-adducts remains very limited. This is not surprising because of the increasingly strong element-element bonds in second and third row group 15 to 17 elements, which make replacement reactions unfavorable. More importantly, a conversion of an (*end*-on bound) ligand into a different ligand coordinated to a nonmetal such as phosphorus, being similar to transition metal ligand transformations, has not yet been described.

Herein, we describe synthesis and exchange reactions of labile ligand-to-phosphinidene complexes as well as the transformation of a nonmetal-bound isocyanide to give either phosphaguanidine or carbene-to-phosphinidene complexes via 1,2-addition of primary amines to the P-C or the C-N bond, respectively. The latter reaction is very much analogous to transformations of transition metal-bound isocyanides. DFT calculations provide insight into the electronic structure as well as reaction mechanisms. The detailed bonding analysis provides an understanding of key NMR features and computational mechanistic studies unveil parallels and differences between nonmetal and transition metal coordination chemistry.

## Results and discussion

We started out from our recent theoretical studies of *end*-on P-ligands of electrophilic terminal phosphinidene complexes, selecting those donors/ligands predicted to have mainly dative ligand-to-P bonds[72]. To probe the concept and to establish the existence of hypothetical transient species, we first investigated the reaction of the Li/Cl phosphinidenoid tungsten complex **1**[61,75] with a set of textbook N-ligands such as pyridine (**2a**), DMAP (**2b**), *N*-MeIm (**2c**) and, in addition, *tert*-butyl isocyanide (**2d**) as important representatives of classical C-ligands from transition metal coordination chemistry. All reactions of **1** with these ligands proceed smoothly in THF to afford the ligand adduct complexes **3a–d** (Fig. 2a) in good yields as yellow solids, except for the case of pyridine (**3a**) which did not go to completion under ambient conditions (content of **3a** at ambient temperature 64% in solution; via ³¹P{¹H} NMR integration). In some cases, small amounts of the common thermal decomposition products of **1** were observed. The isocyanide adduct **3d** was obtained previously in lower yields from 3-iminoazaphosphiridine complexes via an exchange reaction[67]. The signal of the C²H proton of **3c** appears at 6.41 ppm in the ¹H NMR spectrum, hence shifted to high frequency compared to that of the free ligand **2c**[76] due to coordination to the phosphinidene complex. The same effect is also observed for the ¹⁵N{¹H} NMR signals of the adducts that are found at −193.9 ppm (**3b**) and −183.9 ppm (**3c**) for the *P*-bound nitrogen nuclei while they occur at −112 ppm for **2b**[77] and −111.4 ppm for **2c**[78]. This shift to high frequency is similar to those induced by coordination to transition metals where coordination shifts Δδ(¹⁵N) are typically −50 to −100 ppm. For example, −65.3 ppm for pyridine upon coordination in [Pd(phen)(py)(C₂H₅)]CF₃SO₃ (phen = 1,10-phenantroline), in CD₂Cl₂[79,80]. It was exciting that the *N*-MeIm-adduct structure **3c** could also be determined by a single-crystal X-ray diffraction (SC-XRD) structure (Fig. 2b); a related *N*-MeIm-to-phosphorus adduct of [Mo(CO)₅(PPh)] was previously described as a highly reactive intermediate by Mathey and coworkers[70].

The adducts **3b–d** were stable at ambient temperature and all new compounds were fully characterized, except pyridine adduct **3a**, including single crystal X-ray diffraction studies of **3b,c** (**3c**: Fig. 2b; for other molecular structures from SC-XRD see SI-Fig. 111–115). The respective chromium complexes can be obtained analogously (see SI) but for simplicity the following discussion has focused on the more informative W case, in part because of the availability of ¹$J_{W,P}$ NMR

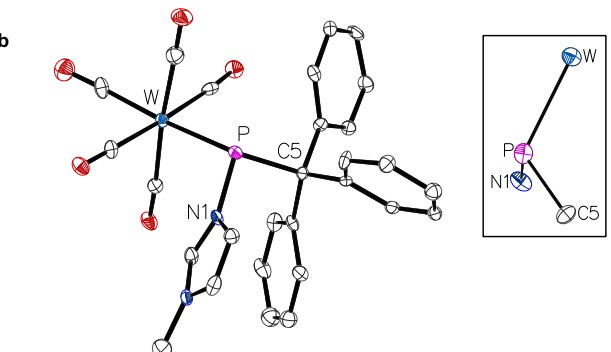

**a**

L = Py (**a**), DMAP (**b**), *N*-MeIm (**c**), tBuNC (**d**)

**b**

**Fig. 2 | Synthesis of ligand-to-phosphinidene complex adducts** 3a–d. **a** Reaction scheme and experimental conditions. **b** Molecular structure of **3c** (left) and the sideview showing the pyramidal geometry of the P center (right). Thermal ellipsoids are set at 50% probability and hydrogen atoms are omitted for clarity. Selected bond lengths / Å and bond angles / °: W-P 2.588(3), P-C5 1.946(10), P-N1 1.813(8), C5-P-W 116.7(3), N1-P-W 108.2(2), N1-P-C5 97.0(4). The isostructural Cr(CO)$_5$ complex is presented in the SI (SI-Fig. 114).

**Table 1 | ³¹P NMR data of the Li/Cl phosphinidenoid complex 1[75], ligand-to-phosphinidene complex adducts 3, phosphaguanidine complexes 11 and the carbene-to-phosphinidene complex adduct 12 as well as calculated [GIAO/CPCM$_{toluene}$/PBE0/def2-TZVP(ecp)//COSMO$_{toluene}$/B3LYP-D3/def2-TZVP] ³¹P NMR chemical shifts of 3a–d**

| Complex | $\delta(^{31}P)$ / ppm (exp.) | $\delta(^{31}P)$ / ppm (calc.) | $^1J_{P,H}$ / Hz | $^1J_{W,P}$ / Hz | Solvent |
|---|---|---|---|---|---|
| **1**[75] | 253.5 | — | — | 76.3 | CDCl$_3$ |
| **3a**[a] | 284.6 | 278.4 | — | <100[b] | THF |
| **3b** | 234.8 | 253.4 | — | 102.5 | THF-d$_8$ |
| **3c** | 199.2 | 254.4 | — | 107.2 | C$_6$D$_6$ |
| **3d** | −50.5[67] | −37.3[c] | — | 117.5 | CDCl$_3$ |
| **11a** | −27.2 | −25.3 | 349.2 | 220.5 | CDCl$_3$ |
| **11b** | −25.8 | −22.3 | 346.9 | 230.9 | C$_6$D$_6$ |
| **12** | −33.8 | — | — | 103.8 | THF |

[a] Data obtained from the reaction solution. [b] ¹⁸³W satellites lie inside the broad signal. [c] Computed for the simplified model complex adduct derived from methylisocyanide (**3d'**).

coupling data. The molecular structure of DMAP-to-phosphinidene complex **3b** was also determined by SC-XRD (SI-Fig. 109) and shows a similar geometry at phosphorus as in **3c** but with a slightly larger bond angle sum at phosphorus ($\Sigma_{\angle P}$ = 324.9(4)° compared to $\Sigma_{\angle P}$ = 321.9(5)° in **3c**). The apparently shorter P-N bond of the *N*-MeIm adduct **3c** (1.813(8) Å) compared to **3b** (1.826(8) Å) is not statistically certain at the 99% confidence level. According to high-level DFT studies (CPCM$_{toluene}$/PWPB95-D3/def2-QZVPP(ecp)), the unusual adduct structure of complexes **3** is also reflected by a significant increase in electron density at phosphorus according to the variation of (Löwdin) electric charges ($\Delta q$ = −0.12, −0.13, −0.16 and −0.08 a.u. for **3a**, **3b**, **3c** and **3d**, respectively) with respect to the electrophilic naked phosphinidene complex **4**, thus reflecting an Umpolung of the terminal phosphinidene complexed P-center. This was confirmed by IR spectroscopic data for complexes **3b–d**, which show increasing CO stretching frequencies for the A$^1$-vibrational mode of the *trans*-CO ligands ($\nu_{CO}$ = 2054, 2053 and 2060 cm$^{-1}$, respectively), thereby consistent with a diminished σ-donation capacity of the phosphorus ligand center in going from **3b** to **3d**.

All complexes **3a–d** show significant ³¹P NMR signatures (Table 1) which are related to some extent to those of Li/X phosphinidenoid complexes[81]. But the comparison also revealed that these belong to two significantly different families of adducts-to-phosphorus with respect to their ³¹P NMR chemical shifts, since the isocyanide adduct **3d** is strongly shielded compared to **1** and **3a–3c**.

The redox chemistry of adducts **3b–d** and the *N*-MeIm-to-phosphinidene chromium(0) complex adduct **3c**$^{Cr}$ was investigated electrochemically by interfacial voltammetry at ceramic screen-printed Pt composite electrodes (Ag/AgCl solid-dot reference electrode) (Fig. 3a). The adducts have a very wide stability range between −2.0 to +0.5 V which, compared to normal organic and transition metal coordination chemistry, is shifted to very negative potentials, i.e. oxidation is facile at these low-coordinate, electron-rich P centers.

The HOMOs of adducts **3b–d** all have a combination of the *p*-lone pair at phosphorus and metal *d* orbital character (**3c,d'**: Fig. 3b). However, while for **3d'** the LUMO has predominantly π*(N≡C) character, the other adducts primarily display π*(W-C) character.

This difference in the redox molecular orbital topologies is mirrored by the cyclic voltammograms, with **3d** differing significantly from the rest. The experimentally obtained peak potentials $E_p^{aI}$ and $E_p^{cIII}$ correlate well with the calculated HOMO-LUMO energies (Table 2), with increasing ease of oxidation (**3c**$^{Cr}$ < **3b** < **3c** « **3d**) for decreasing HOMO energy (**3b** > **3c** > **3d**), and increasing ease of reduction (**3d** < **3b** < **3c**$^{Cr}$ < **3c**) for increasing LUMO energy (**3d** < **3b** < **3c**).

The stability of the P-adducts **3c,d** was probed with a VT ³¹P{¹H} NMR study to examine the thermal P-ligand dissociation and, hence, also the concomitant formation of the free terminal phosphinidene complex **4** (Fig. 4a). Complex **3c** was stable in the VT NMR experiment up to 60 °C, but with further temperature increases (70 to 100 °C), the intensity of the signal of **3c** started to decrease, while only the formation of white phosphorus (**8**) could be observed; unfortunately, no evidence for formation of **4** was obtained. However, full conversion to **8** was only achieved by heating the reaction mixture at 90 °C for 16 hours. When complex **3d** was heated to 100 °C, the decomposition to complex **9** was observed (86% in solution; by ³¹P NMR integration next to various side products, inter alia white phosphorus (**8**) at −520.6 ppm, 1% in solution). Upon changing the metal center from tungsten to chromium, using the same protocol, the isocyanide-to-phosphinidene complex adduct **3**$^{Cr}$**d** was observed to decompose selectively to give **8** at 85 °C. However, the respective *N*-MeIm P-adduct **3**$^{Cr}$**c** showed an additional product shifted to lower frequency at 569.4 ppm in the ³¹P{¹H} NMR spectrum in the range of 70 to 90 °C that appeared to be an intermediate, before the starting material fully decomposed to yield **8** at 100 °C (Fig. 4a). The ³¹P NMR chemical shift of this intermediate fits well to the bis(triphenylmethyl)diphosphene (**6**), which was previously reported by Schmutzler[82]. Furthermore, ³¹P{¹H} NMR evidence for the butterfly-type P$_4$ derivative **7** was obtained in the thermolysis using chromium complexes **3**$^{Cr}$**b** and **3**$^{Cr}$**c**. Formation of the DMAP tungsten complex **5b** resulted under mild heating but was also detected under mass spectrometric conditions using the liquid injection field desorption ionization (LIFDI) technique.

These preliminary results on the P-ligand bond strength prompted us to study ligand substitution reactions using the *N*-MeIm adduct **3c** as the most meaningful example to investigate. Adding DMAP (**2b**) or *tert*-butyl isocyanide (**2d**) to a THF solution of **3c** led to a very fast ligand exchange, even at ambient temperature, thus yielding the P-adduct complexes **3b,d** selectively (Fig. 4b). Unfortunately, no reaction was observed with carbon monoxide at ambient temperature to form the CO adduct (**3e**). Upon heating to 70 °C a very unselective reaction was observed, thus leading to a complicated product mixture

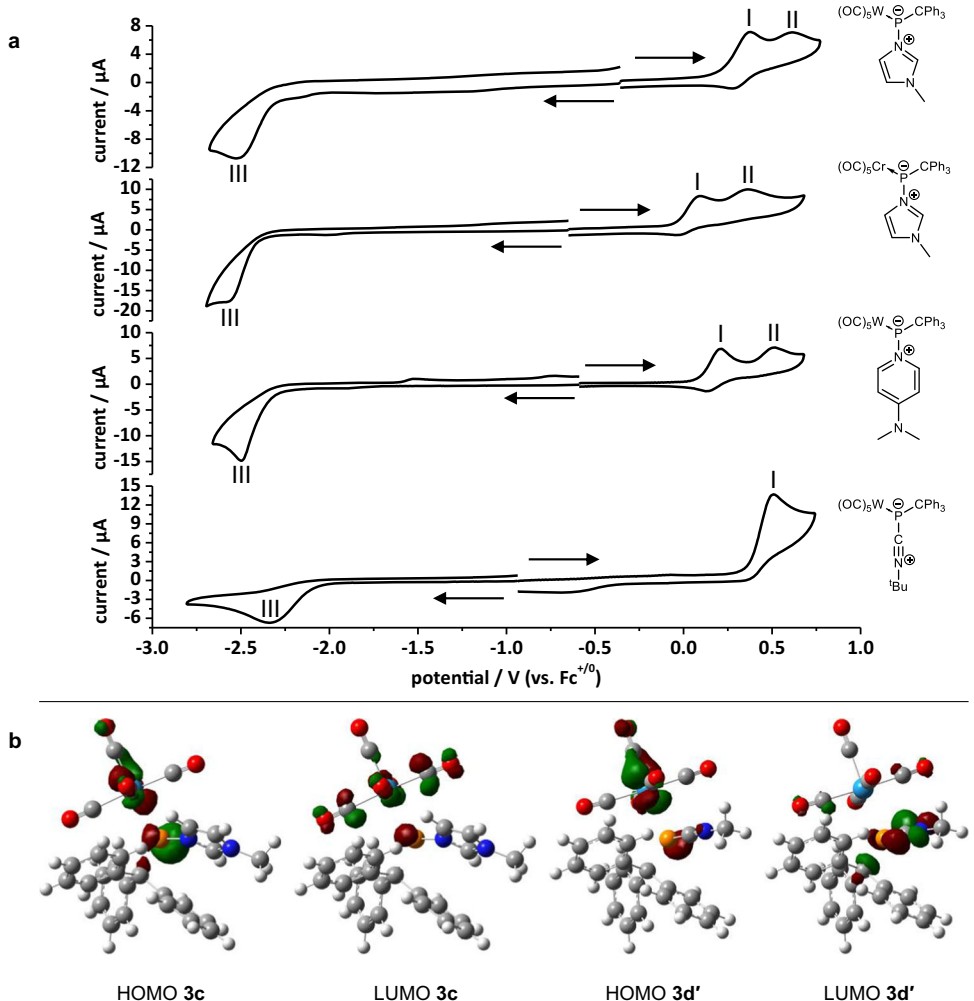

**Fig. 3 | Electrochemical investigation. a** Overlay of cyclic voltammograms of **3b**, **3c, 3c^Cr** and **3d** at a Pt electrode in a 0.2 M $^n$Bu$_4$PF$_6$/THF solution; analyte concentration of 0.5 mM (**3b**) and 1.0 mM (**3c, 3c^Cr, 3d**) oxidation parts with anodic scan direction and reduction parts with cathodic initial scan direction as denoted with arrows; scan rate: 200 mV s$^{-1}$; potentials are referenced versus Fc$^{+/0}$. **b** Plots of the Kohn-Sham HOMO and LUMO of complexes **3c** and **3d'**.

consisting of a multitude of products, several exhibiting AB-type spin systems, in the $^{31}$P{$^1$H} NMR spectrum besides other products observed in the case of **2b, 2d**, or in the thermal decomposition of **3c** (see also Table 3, and the SI-Fig 40–43). To tackle the intriguing question of the aforementioned Umpolung at phosphorus upon ligand addition the reactivity towards non-activated, non-polarized alkenes was examined (Fig. 4b). Therefore, a solution of **3c** was treated with 1-pentene and 1-hexene and followed by $^{31}$P NMR spectroscopy. At ambient temperature no reaction was observed, but upon warming to 70 °C, suddenly, the selective formation of a diastereomeric mixture of phosphirane complexes **10a** (34:66) and **10b** and (36:64) with $^{31}$P NMR chemical shifts of −137.7 ppm (**cis−10a,** $^1J_{W,P}$ = 258.8 Hz) and −145.4 ppm (**trans−10a,** $^2J_{P,H}$ = 19.6 Hz, $^1J_{W,P}$ = 259.7 Hz), and −137.4 ppm (**cis−10b,** $^1J_{W,P}$ = 258.9 Hz) and −145.5 ppm (**trans−10b,** $^2J_{P,H}$ = 15.2 Hz, $^1J_{W,P}$ = 260.2 Hz) was observed, indicating a ligand dissociation prior to the rapid [2 + 1] cycloaddition of the transient terminal phosphinidene complex **4**. At this point it should be also noted that π-systems such as alkenes can coordinate reversibly to phosphinidene complexes to form respective phosphirane complexes if terminal alkenes are used as solvent[83,84].

One of the cornerstone findings in transition metal organometallic chemistry is the addition of protic nucleophiles such as amines or alcohols to coordinated isocyanide to form carbene metal complexes as described by Richards and others[18–20]. To test whether a P-bound ligand in complex **3d** could behave similarly to a transition metal-bound ligand, the following reactions with primary amines were investigated.

When the isocyanide-to-P complex **3d** was treated with methylamine in THF, a selective conversion gave the η$^1$-phosphaguanidine-κ*P* complex **11a** at ambient temperature (Fig. 5a). The phosphorus chemical shift of **11a** (−27.2 ppm) fits very well with the rare η$^1$-phosphaguanidine complex [W(CO)$_2$(η$^5$-C$_5$H$_5$){Ph$_2$PC(N-*p*-tol)N(H)*p*-tol}Cl] (δ($^{31}$P) = − 25.6 ppm)[85]. The structure of **11a** was confirmed by single crystal X-ray crystallography (Fig. 5b). When **3d** was treated with isopropylamine at ambient temperature, the reaction proceeded much more slowly than with methylamine and full conversion was observed only after one week, but the outcome was again the 1,2-addition to the P-C(ligand) bond. The phosphaguanidine complex **11b** (−25.8 ppm) was formed as 1,2-addition product of the N-H to the P-C bond in **3d** (Fig. 5a and Table 1), including a surprising prototropic rearrangement. When the reaction was performed in neat isopropylamine, full conversion was completed within one day.

To test the selectivity of the addition reaction to **3d**, a further increase in steric demand of the amine R substituent (R = $^t$Bu) was chosen, while keeping all reaction conditions constant. In the case of **3d**, the reaction was drastically slowed so that a full conversion to the product **12** was not achieved at ambient temperature. Raising the reaction temperature to 40 °C did not lead to full conversion either

and above 40 °C the reaction became increasingly unselective. The $^{31}$P NMR spectrum of the reaction solution showed a singlet for **12** and a value of −33.8 ppm which is slightly shifted to higher frequency compared to those of **11a,b**, but without any P,H coupling. The drastically smaller $^1J_{W,P}$ coupling constant of 103.8 Hz in **12** (compared to **11a,b**) indicates an increased negative charge density at phosphorus, similar to the zwitterionic complexes **3a–d** but with a significantly different chemical shift. The resulting 1,2-C-N addition product **12** represents an example of an isocyanide-to-carbene ligand conversion at a neutral nonmetal center, analogous to transition metal coordination chemistry[21–23].

Computed $^{31}$P NMR chemical shifts (see Computational Details in the Theoretical Investigations section of the SI) for the model phosphaguanidine complexes **16a$^p$** (−25.3 ppm) and **16 b$^p$** (−22.3 ppm) (vide infra) are in very good agreement with the aforementioned shift to low frequency observed for the experimentally obtained product in the reaction with isopropylamine **11b**, compared to that of methylamine

**11a** (Table 1). These data strongly suggested the formation of the zwitterionic heteroatom-substituted carbene-to-phosphinidene adduct **12** in the reaction with *tert*-butylamine, which is a valence isomer of the hypothetical phosphaalkene complex **13** (Fig. 5a), the former having a different non-classical geometry and hybridization at the phosphorus center in **12**. Early examples of so-called inversely polarized phosphaalkene complexes, having two amino substituents at the carbon atom, have comparable $^{31}$P NMR data, e.g. [W(CO)$_5${$^t$BuP = C(NMe$_2$)$_2$}]  (δ($^{31}$P) = −25.1 ppm,  $^1J_{W,P}$ = 153.5 Hz)[86] or [W(CO)$_5${PhP(IDipp)}] (IDipp = 1,3-bis(2,6-diisopropylphenyl)imidazol-2-ylidene) (δ($^{31}$P) = − 57.7 ppm, $^1J_{W,P}$ = 120 Hz)[87].

For a better understanding of the reactivity of the electrophilic phosphorus center both in ligand substitution reactions at the P atom and in the formation of metal ligated phosphiranes, quantum chemical calculations at the CCSD(T)/def2-TZVPP(ecp) level (see the SI for computational details[88]) were performed for two model reactions of the most labile *N*-methylimidazole adduct and using the simplified *P*-*tert*-butyl substituted pentacarbonyl-tungsten(0) complex model **3c″** for the sake of computational efficiency (Fig. 6a). Ligand exchange at P with (model) MeNC ligand takes place exergonically by an $S_N2$ associative mechanism with a rather low barrier. However, reaction of **3c″** with ethene as the model olefin requires the initial, endergonic barrierless dissociation of the *N*-MeIm ligand, forming model phosphinidene **4″** as a true intermediate, followed by barrierless chelotropic cycloaddition affording the model phosphirane complex **10c″** in an overall exergonic process (Fig. 6a). For the latter step, a low barrier was found previously in the case of *P*-amino-substituted phosphinidene complexes[89]. Complex **14** could also be formed by a direct dipolar cycloaddition reaction of **3c″** with ethene, although this is unfavorable due to its slightly lower stability and the higher energy barrier of this step. Indeed, such a bicyclic derivative has never been observed. Interestingly, almost identical values were obtained at the much faster PBEh-3c optimization level (see the SI).

To understand the coordination vs. transition metal-like reactivity of the isocyanide *end*-on adduct **3d** towards primary amines, i.e., the

**Table 2 | Cyclic voltammetry correlated to [CPCM$_{toluene}$/ PWPB95/def2-QZVPP(ecp)] redox molecular orbital energies $E_P^a$ and $E_P^c$ versus Fc$^{+/0}$.[a]**

| Complex | $E_P^{aI}$ / V | $E_P^{aII}$ / V | HOMO / eV | $E_P^{cIII}$ / V | LUMO / eV | ΔE / V[b] | H-L / eV |
|---------|--------|--------|-----------|--------|-----------|-------|---------|
| **3b** | −0.04 | +0.17 | −5.03 | −2.58 | −1.54 | 2.54 | 3.49 |
| **3c** | −0.02 | +0.20 | −5.20 | −3.05 | −1.33 | 3.03 | 3.88 |
| **3c$^{Cr}$** | −0.07 | +0.40 | — | −2.95 | — | 2.87 | — |
| **3d** | +0.51 | — | −5.53[c] | −2.34 | −1.81[c] | 2.85 | 3.72[c] |

[a] *Fc* in this paper designates ferrocene; [Fe(η$^5$-C$_6$H$_5$)$_2$] and the ferrocene/ferrocenium redox couple is designated as Fc$^{+/0}$, set to 0 V according to IUPAC recommendations[102]. $E_P^a$ and $E_P^c$ designate anodic and cathodic peak potentials, respectively, in standard electrochemical notation. [b] ΔE = $E_P^{aI} − E_P^{cIII}$. [c] Computed for the simplified model complex adduct derived from methylisocyanide (**3d′**).

**Fig. 4 | Experimental study on the P-donor bond strength. a** Dissociation of ligand-to-phosphinidene complex adducts **3c,d** and **3$^{cr}$c,d. b** Ligand substitution reactions of phosphinidene complex adducts.

**Table 3 | Computed [CPCM$_{tol}$/PWPB95-D3/def2-QZVPP(ecp)//CPCM$_{tol}$/B3LYP-D3/def2-TZVP(ecp)] energetic, electronic and bond-strength related parameters for 3a–c,d′ and 12**

| Complex | Ligand | ΔG$_{comp}$ [a] | BDE$_{P-L}$ [a] | E$_{def}$ [a] | ε$_{HOMO}$ / eV | q$^{Mull}_{phos}$ / e | WBI$_{L-P}$ |
|---|---|---|---|---|---|---|---|
| **3a** | Py | −9.94 | 26.40 | 14.86 | −6.12 | −0.45 | 0.628 |
| **3b** | DMAP | −15.30 | 32.16 | 15.26 | −5.92 | −0.51 | 0.622 |
| **3c** | *N*-MeIm | −13.26 | 30.63 | 15.79 | −6.07 | −0.51 | 0.628 |
| **3d′** | CNMe | −21.44 | 37.24 | 13.86 | −6.34 | −0.33 | 1.244 |
| **12** | C(NH$^t$Bu)$_2$ | −36.53 | 55.65 | 24.05 | −5.90 | −0.55 | 1.046 |
| **3e** | CO | −28.16 | 23.52 | 13.35 | −6.69 | −0.08 | 1.426 |

[a]In kcal·mol⁻¹; here the heterolytic BDE values are given, for more details see reference[103].

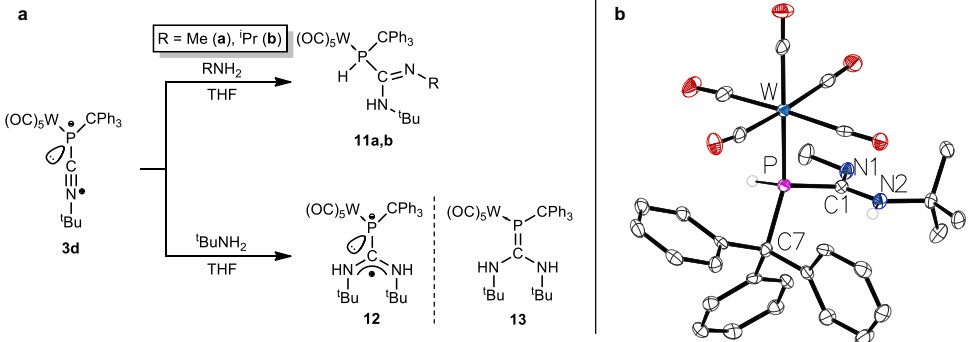

**Fig. 5 | 1,2-Addition reactions of amines. a** Reaction scheme for the amine addition (i) to the P-C and (ii) C-N bond of complex **3d**. **b** Molecular structure of **11a**. Thermal ellipsoids are set at 50% probability and hydrogen atoms are omitted for clarity except those bound to phosphorus and nitrogen atoms. Selected bond lengths / Å and bond angles / ° (bond lengths and bond angles in square brackets are of the second independent molecule of **11a**): W-P 2.5166(12) [2.5239(12)], P-C7 1.931(5) [1.915(5)], P-C1 1.867(5) [1.889(5)], N1-C1 1.272(6) [1.270(6)], N2-C1 1.371(6) [1.363(6)], C1-N1-C2 121.9(4) [123.2(4)], C1-N2-C3 126.3(4) [126.7(4)], N1-C1-P 125.5(4) [125.3(4)], N2-C1-P 112.2(3) [112.5(3)], N1-C1-N2 121.7(4) [121.4(4)]. The homologous Cr(CO)$_5$ complex is presented in the SI (SI-Fig. 117).

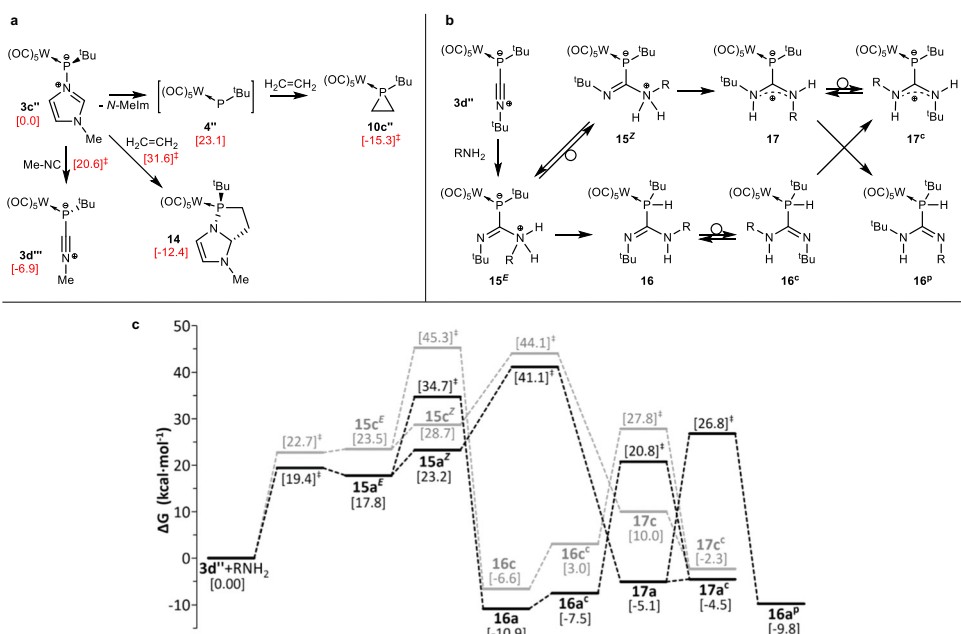

**Fig. 6 | Computed proposed reaction mechanisms. a** Proposed mechanism for the reactions of model *N*-MeIm adduct complex **3c″** with methyl isocyanide and ethene. Computed [CPCM$_{toluene}$/CCSD(T)/def2-TZVPP(ecp)//CPCM$_{toluene}$/B3LYP-D3/def2-TZVP(ecp)] relative Gibbs free energies (kcal·mol⁻¹) in red and square brackets. **b** Proposed mechanism for the reaction of model isocyanide adduct complex **3d″** with alkyl amines. **c** Computed [COSMO$_{THF}$/CCSD(T)/def2-TZVPP(ecp)//COSMO$_{THF}$/B3LYP-D3/def2-TZVP(ecp)] relative Gibbs free energy profile for the reaction of model isocyanide adduct complex **3d″** with methyl (a in black) or *tert*-butyl amine (c in grey).

competing 1,2-addition to the P-C(ligand) and the C-N bond, various aspects of bonding were closely inspected by theoretical means at the same level and using the simplified *P*-tert-butyl substituted derivative **3d″** derived from *P*-tert-butyl isocyanide (Fig. 6b). The nucleophilic attack of methylamine to the highly electrophilic isocyanide ligand C atom furnishes intermediate **15a^E** (a stands for R = Me, whereas *E/Z* refers to the imine group configuration) that readily undergoes a [1,3]H shift over a moderate barrier to the very stable phosphaguanidine complex **16a**. Rotamer **16a^c** (superscript c stands for a conformer) facilitates the kinetically hampered proton-transfer from phosphorus to the other N atom originally belonging to the isocyanide ligand, giving rise to a conformer (**17a^c**) of the somewhat more stable diaminocarbene **17a**. The latter could also be formed by N-to-N proton transfer in the *Z*-configured isomer **15a^Z**, although this route is disfavored compared to the above mentioned prototropy in **15a^E**. The experimentally observed product **11a** in the reaction with methylamine (Fig. 5a) does not have exactly the same structure as the expected model compound **16a** or **16a^c**, but is a protomer of the latter, (denoted with the p superscript) **16a^p**, arising either via intermolecular (not studied) or intramolecular proton transfer from **17a**. The sterically more demanding isopropylamine gives rise to similar or slightly higher energy model intermediates (not shown) **15b^E** ($\Delta G = 20.5$ kcal·mol$^{-1}$), **16b** ($\Delta G = -10.9$ kcal·mol$^{-1}$) and the most stable final product **16 b^p** ($\Delta G = -12.6$ kcal·mol$^{-1}$). This seems to indicate that the increase of steric bulk favors **16^p** over the initially formed intermediate **16** and, hence, points to a thermodynamic preference for the protomer also in case of reaction with methylamine when the true, more sterically crowded, triphenylmethyl group is used as *P*-substituent (**3d**).

As expected, the bulkier *tert*-butylamine gives rise to a higher energy profile compared to methylamine (Fig. 6c). The most significant difference is the lower barrier for the N-to-N proton transfer **15c^Z → 17c** ($\Delta\Delta G^{\ddagger} = 15.4$ kcal·mol$^{-1}$) compared to the N-to-P one **15c^E → 17c** ($\Delta\Delta G^{\ddagger} = 21.8$ kcal·mol$^{-1}$), which, after rotation, makes the diaminocarbene-adduct conformer **17c^c** the kinetically preferred product. This agrees with the experimental (³¹P NMR) observation in case of the reaction with *tert*-butylamine. The proposed zwitterionic structure for experimental (**12**) or model (**17/17^c**) diaminocarbene-phosphinidene complex is supported by the similar values obtained for C-P bond strength parameters in **17c^c** compared to a reported model NHC-phosphinidene complex[73]. ($d = 1.831/1.819$ Å; WBI = 1.059/1.020; MBO = 1.036/1.071; $\rho(r) = 0.1487/0.1473$ au; $\frac{1}{4}\nabla^2\rho(r) = -0.0473/-0.0180$ e·Å$^{-5}$). Indeed, the electron transfer from the ligand to the phosphinidene complex fragment in model diaminocarbene-phosphinidene adduct **17c^c** ($\Delta q^{\text{Löwdin}} = -0.35$ e) is much higher than the values reported beforehand for **3a–d**. Therefore, it indicates that the above-mentioned P-C(NH^tBu)$_2$ linkage in **17c^c** should be better considered as dative C → P bonding. It should be noted here that the calculations not only support the experimental results but also clearly reveal differences between the chemistry of isocyanide metal complexes and the P-adducts reported here. In the former case a 1,2-addition reactivity to the M-C(ligand) bond does not exist, while in case of the P-C(ligand) bond it is the preferred pathway for small primary amines. Only if sterics become important does the organometallic-like reactivity of the phosphorus center comes to the fore, i.e., the isocyanide-to-carbene ligand conversion via 1,2-addition to the C-N bond.

As recently pointed out[90] from their thermodynamic oxygen-transfer potentials (TOP)[91], organyl-substituted phosphinidene κ*P*-pentacarbonyltungsten(0) complexes, R-P-W(CO)$_5$, possess highly electrophilic phosphorus centers. Consequently, they are more easily oxidized to give the respective terminal phosphinidene oxide complex as in case of the *P*-methyl-substituted phosphinidene complex (TOP = $-414$ kJ·mol$^{-1}$ = $-98.9$ kcal·mol$^{-1}$), compared to ethyl isocyanide (TOP = $-395$ kJ·mol$^{-1}$ = $-94.4$ kcal·mol$^{-1}$). At the much higher level of theory PWPB95-D3/def2-QZVPP(ecp)//RIJCOSX-

B3LYP-D3/def2-TZVP(ecp), the computed TOP for R-P-W(CO)$_5$ increase in the order CPh$_3$ < Me < $^t$Bu ($-438$, $-473$, $-480$ kJ·mol$^{-1}$; or $-104.8$, $-113.2$, $-114.8$ kcal·mol$^{-1}$) (SI-Fig. 121).

The fluoride ion affinity (FIA)[92–94], defined as the negative enthalpy change of the gas-phase reaction of an acid A to form the adduct [A-F]$^-$, has been used as benchmark for the quantification of Lewis acidity for many different species including phosphorus centers such as phosphenium cations[95]. According to the FIA, the (hard) acidity of phosphinidene complexes R-P-W(CO)$_5$ increases in the order CPh$_3$ < Me < $^t$Bu (400.5, 407.6 and 414.2 kJ·mol$^{-1}$; or 95.8, 97.5 and 99.1 kcal·mol$^{-1}$, respectively) (SI-Fig. 121). For comparison, FIA values (kJ·mol$^{-1}$) were also computed at the working level of theory (see Computational Details at the SI) for common reference compounds BF$_3$ (355.0), AlF$_3$ (491.5), SiF$_4$ (318.5), PF$_5$ (390.8) and SbF$_5$ (498.0), in good agreement with reported values[94] (345.1, 483.6, 311.1, 382.0 and 495.1, respectively) computed at the CCSD(T)/CBS//PBEh-3c level. It should be noted here that the effect of low-coordinate element centers has not yet been thoroughly studied, but it seems to be apparent that the FIA value always increases with the coordination number/oxidation state[94].

Moreover, additional evidence of the (varying) dative character for all herein described ligand-phosphinidene complex adducts **3a** (instead of **3d** with L = $^t$BuNC, the simplest methylisocyanide complex adduct **3d′** was used) and **12**, is provided using Bader's quantum theory of atoms in molecules (QTAIM). The Holthausen-Cowley criteria for dative bonding are fulfilled by the L → P bonds of adduct complexes **3a–c** and **12** (Fig. 7a):

i. existence of a bond critical point (BCP) closer to the electron acceptor center (P in these cases)
ii. with vanishing $\nabla^2\rho$ value
iii. displaying two valence-shell charge concentration (VSCC) regions (bands) located at the basin of the donor atom along the central part of the bond path[96,97].

Most recently, the relative charge concentration bands position parameter, $\tau_{\text{VSCC}}$, together with the small positive value of $\nabla^2\rho$ at the BCP, was reported to allow unequivocal assignment to dative bonding[73]. The parameter $\tau_{\text{VSCC}}$ is defined as the product of the two (signed) VSCC positions, divided by the square of the bond path distance (to provide a non-dimensional quantity), here $\tau_{\text{VSCC}} = 0.0013$. The positions of the minima for VSCC$_{N/C}$ and VSCC$_P$ (e.g., $-0.624$ and $-0.007$ Å, respectively, for **12**) were obtained by deconvolution of the central part of the $\nabla^2\rho$ plot into asymmetric Gaussian functions, as previously reported[73].

Furthermore, adduct complexes **3a–c** and **12** also show significant electron density transfers from the ligands to the phosphinidene complex units of ca. 0.5 *e*, and relatively high HOMO energy values above $-6.12$ eV (Table 3). The adduct resulting from a carbene donor (**12**) exhibits a robust single C-P bond with a WBI close to unity and a rather large Gibbs free energy of complexation, whereas those arising from heteroaromatic N-donors (**3a–c**) display moderately weak N-P bonds and rather low complexation energies. Among them, **3b** represents an interesting case because it constitutes the most labile isolable adduct (**3a** was not easily isolable in pure form) thus being the optimal candidate for ligand substitution reactions (vide infra). The most different situation is that observed for **3d′** (Fig. 7a), which features a remarkably high $\nabla^2\rho$ value (7.87 e·Å$^{-5}$) that indicates mostly covalent character for the ligand-P bond. This is in line with low electron density transfer to the phosphinidene complex unit (0.33 e), because of π-backdonation to the isocyanide ligand, which is supported by the low-lying HOMO and the partial double-bond character, as indicated by the high WBI (Table 3). However, **3d′** keeps a highly pyramidalized geometry at P ($\Sigma_{\angle P} = 320.3°$) and its valence tautomer with a planar P center corresponds to the *vertex*-type TS for the inversion at P, with a remarkably low barrier ($\Delta\Delta G^{\ddagger} = 2.8$ kcal·mol$^{-1}$) compared to the other three-coordinated P(III) species[98]. The natural resonance theory (NRT) analysis of a very simplified model **3d^{Me,Me}** with

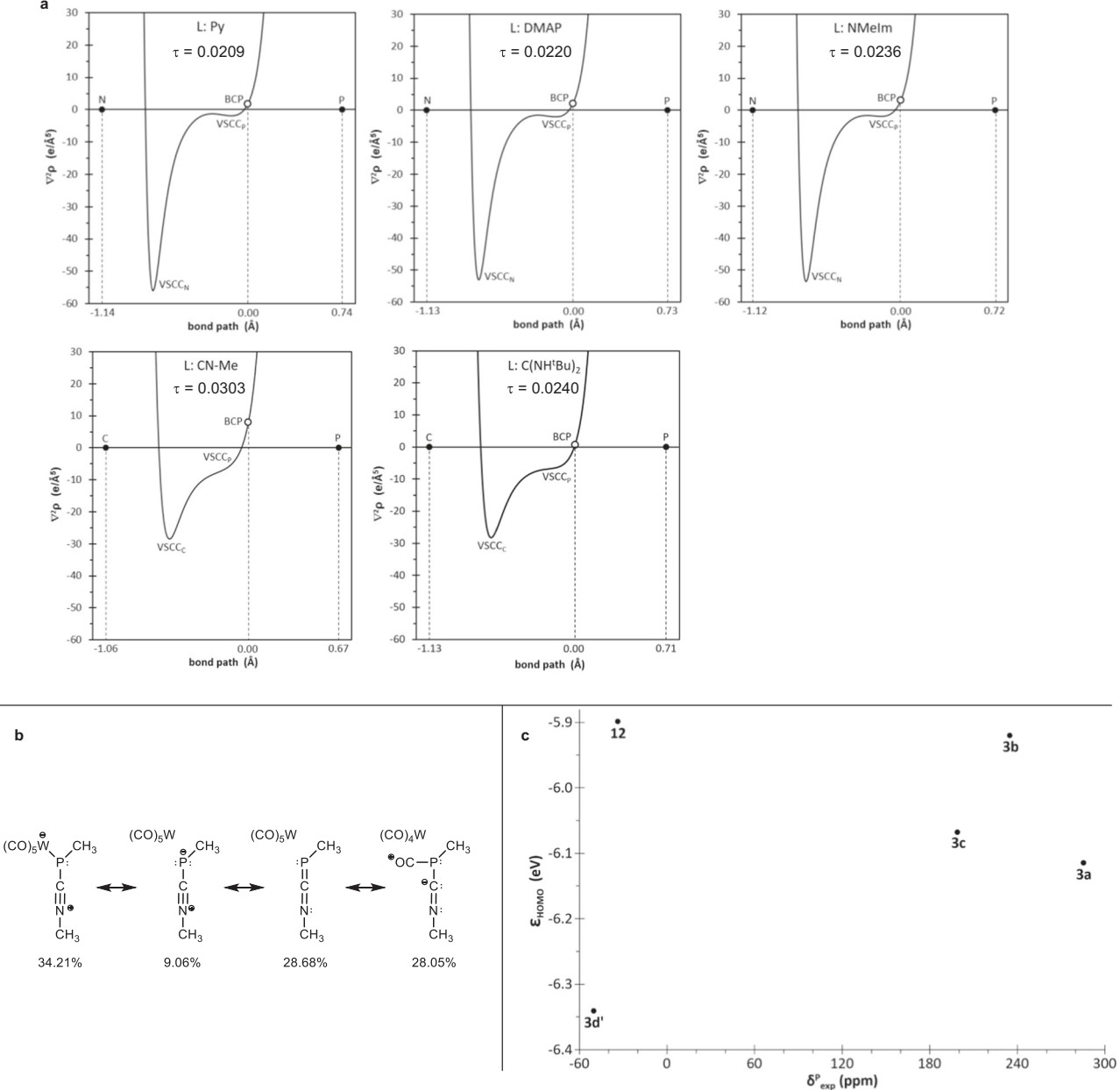

**Fig. 7 | Bonding analysis and electronic properties. a** Computed [B3LYP-D3/def2-TZVPP(ecp)//B3LYP-D3/def2-TZVP(ecp)] variation of the Laplacian of electron density $\nabla^2\rho$ for **3a–c,d'** and **12** along the L-P bond path. **b** NRT analysis of model adduct **3d^{Me,Me}**. **c** Plot of the computed [COSMO$_{THF}$/CCSD(T)/def2-TZVPP(ecp)//COSMO$_{THF}$/B3LYP-D3/def2-TZVP(ecp)] HOMO energies with the experimental $^{31}$P chemical shift values.

methylisocyanide as ligand and a *P*-methyl substituent, reveals that the isocyanide ligand is linked by a formal P = C in a 28.7% extent, considering the twelve resonance structures contributing more than 2.1% (Fig. 7b).

However, the expected correlation between the HOMO energies and the experimental $^{31}$P NMR chemical shift values can only be roughly observed if the isocyanide adduct **3d'** is not considered (Fig. 7c; for a correlation with HOMO-LUMO gaps see SI-Fig. 120).

Worth mentioning is that the reaction pathway of **3c** with CO was not studied here because the small heterolytic BDE value (Table 3) and the unselective reaction at elevated temperatures indicated that the CO-to-P adduct formation is not favored to yield a stable product.

We have demonstrated that a neutral, low-coordinate nonmetal element center, i.e., phosphorus, can undergo substitution and ligand conversions reminiscent of transition metal reactions. The case of primary amines also clearly reveals differences to metal-bound ligand

conversions as a 1,2-addition of the N-H bond to an M-C bond of an isocyanide ligand is not preferred, while in case of phosphorus it is for less-encumbered amines. If sterics become more important, then the parallel in reactivity comes to the fore, the 1,2-addition to the C-N bond. The cyclic voltammetry studies revealed a very wide stability range which is shifted to negative potentials compared to common organic and transition metal coordination chemistry. The phosphorus-ligand bond is properly described as dative bonding for heteroaromatic N-donors and diaminocarbene ligands, according to the relative charge concentration bands position and the Laplacian of the electron density at the BCP. However, isocyanide ligands are bound through a linkage with partial double bond character, featuring a low-lying HOMO and low net electron density transfer from ligand to P fragment, due to π-backdonation. Overall, these results not only complete a jigsaw puzzle that had already started to appear in the late 1990s with the chemistry of transiently formed, but not observed species then

called nitrilium phosphane-ylid complexes reacting as 1,3-dipoles[99–101]. We believe that our findings will not only inspire and open new avenues in nonmetal chemistry but also may lead to re-evaluation of older observations.

## Methods

### Synthesis of complex 3b

A solution of a Li/Cl phosphinidenoid tungsten(0) complex **1** was prepared using 0.078 g (0.12 mmol, 1.0 eq.) of [W(CO)$_5${P(CPh$_3$)Cl$_2$}], 18 µL (0.11 mmol, 1.0 eq.) of 12-crown-4 and 0.08 mL ($c$ = 1.6 M in $n$-pentane, 0.13 mmol, 1.1 eq.) of a *tert*-butyllithium solution in 2 mL of THF at −80 °C. Afterwards, 0.030 g (0.24 mmol, 2.1 eq.) of 4-dimethylaminopyridine (DMAP) (**2b**) was added at −50 °C. The reaction mixture was stirred for 18 h while it was allowed to slowly warm up to ambient temperature. All volatiles were removed in vacuo (<0.02 mbar) at ambient temperature. The product was extracted five times with 5 mL of diethyl ether using a filter cannula (∅ = 2 mm) with a glass microfiber filter paper (Whatman® GF/B) and a Schlenk frit (filled with dry SiO$_2$, ∅ = 1 cm, $h$ = 2 cm). Afterwards, 40 mL of $n$-pentane were added to the solution. The obtained yellow suspension was stirred for 1 h at ambient temperature. The supernatant was filtered off using a filter cannula (∅ = 2 mm) with a Whatman® 595 filter paper at ambient temperature and the yellow solid residue was washed three times using 5 mL of $n$-pentane at ambient temperature. The product was obtained as yellow solid after drying for 1 h in vacuo (<0.02 mbar) at ambient temperature. Yield: 0.045 g (0.06 mmol, 54%); yellow solid; mp: 141 °C (dec.); $^1$H NMR (500.04 MHz, THF-$d_8$, 298 K): δ 7.65 (d, $J$ = 6.7 Hz, 2H), 7.57−7.55 (m, 6H), 7.18−7.14 (m, 6H), 7.09−7.05 (m, 3H), 6.47−6.46 (m, 2H), 3.06 (s$_{sat}$, $J$ = 75.25 Hz, 6H); $^{13}$C{$^1$H} NMR (125.75 MHz, THF-$d_8$, 298 K): δ 203.5 (d, $J$ = 13.6 Hz), 200.4 (d$_{sat}$, $J$ = 7.6 Hz, $J$ = 126.1 Hz), 156.8, 151.5 (d, $J$ = 10.7 Hz), 147.4, 130.9 (d, $J$ = 11.5 Hz), 128.3, 125.9, 106.5, 63.9 (d, $J$ = 53.4 Hz), 39.4; $^{15}$N{$^1$H} NMR (50.68 MHz, THF-$d_8$, 298 K): δ −193.9, −298.3; $^{31}$P NMR (202.44 MHz, THF-$d_8$, 298 K): δ 234.8 (br s$_{sat}$, $J$ = 102.5 Hz); IR (ATR Diamond): 1917 (ν$_{CO}$), 1971 (ν$_{CO}$), 2054 (ν$_{CO}$); MS (EI, 70 eV): $m/z$ (%) = 598.0 (<1) [W(CO)$_5$(PCPh$_3$)]$^+$, 569.9 (<1) [W(CO)$_4$(PCPh$_3$)]$^+$, 541.9 (<1) [W(CO)$_3$(PCPh$_3$)]$^+$, 514.0 (<1) [W(CO)$_2$(PCPh$_3$)]$^+$, 274.0 (<1) [PCPh$_3$]$^+$, 243.0 (100) [CPh$_3$]$^+$, 121.0 (48) [DMAP-H]$^+$; MS (LIFDI): $m/z$ (%) = 721.2 (38) [$M$ + H]$^+$, 446.0 (10) [W(CO)$_5$(dmap)]$^+$, 243.2 (22) [CPh$_3$]$^+$; elemental analysis calcd (%) for C$_{31}$H$_{25}$N$_2$O$_5$PW: C 51.69, H 3.50, N 3.89; found: C 51.51, H 3.76, N 3.74.

### Synthesis of complex 3c

A solution of a Li/Cl phosphinidenoid tungsten(0) complex **1** was prepared using 1.425 g (2.13 mmol, 1.0 eq.) of [W(CO)$_5${P(CPh$_3$)Cl$_2$}], 0.34 mL (2.10 mmol, 1.0 eq.) of 12-crown-4 and 1.34 mL ($c$ = 1.6 M in $n$-pentane, 2.14 mmol, 1.0 eq.) of a *tert*-butyllithium solution in 22 mL of THF at −80 °C. Afterwards, 0.26 mL (3.26 mmol, 1.5 eq.) of $N$-methylimidazole (**2c**) were added dropwise at −50 °C. The reaction mixture was stirred for 15.5 h while it was allowed to slowly warm up to ambient temperature. All volatiles were removed in vacuo (<0.02 mbar) at ambient temperature. The product was extracted six times with 30 mL of diethyl ether using a filter cannula (∅ = 2 mm) with a glass microfiber filter paper (Whatman® GF/B) and a P3 Schlenk frit (filled with dry SiO$_2$, ∅ = 3 cm, $h$ = 3 cm). Residual product was extracted from the SiO$_2$ using three times 40 mL of diethyl ether. Afterwards, 420 mL of $n$-pentane were added to the solution. The obtained yellow suspension was stirred for 1 h at ambient temperature. The supernatant was filtered off using a filter cannula (∅ = 2 mm) with a Whatman® 595 filter paper at ambient temperature and the yellow solid residue was washed three times using 20 mL of $n$-pentane at ambient temperature. The product was obtained as yellow solid after drying for 9 h in vacuo (<0.02 mbar) at ambient temperature. Yield: 0.805 g (1.18 mmol, 55%); yellow solid; mp: 146 °C (dec.); $^1$H NMR (300.13 MHz, C$_6$D$_6$, 298 K): δ 7.75−7.73 (m, 6H), 7.14−7.09 (m, 6H), 7.00−6.94 (m, 3H), 6.41−6.40 (m, 1H), 6.33−6.32 (m, 1H), 5.11−5.10 (m, 1H), 1.83 (s, 3H); $^{13}$C{$^1$H} NMR

(125.75 MHz, C$_6$D$_6$, 298 K): δ 203.2 (d, $J$ = 14.0 Hz), 200.4 (d$_{sat}$, $J$ = 7.4 Hz, $J$ = 126.2 Hz), 147.4 (d, $J$ = 8.4 Hz), 142.1, 130.5 (br s), 130.3 (d, $J$ = 7.9 Hz), 128.4, 125.9, 119.8, 62.9 (d, $J$ = 49.9 Hz), 33.9; $^{15}$N{$^1$H} NMR (50.68 MHz, C$_6$D$_6$, 298 K): δ −183.9, −215.2; $^{31}$P NMR (121.51 MHz, C$_6$D$_6$, 298 K): δ 199.2 (br s$_{sat}$, $J$ = 107.2 Hz); IR (ATR Diamond): 1921 (ν$_{CO}$), 1966 (ν$_{CO}$), 2053 (ν$_{CO}$); MS (LIFDI): $m/z$ (%) = 680.8 (38) [$M$]$^+$, 243.1 (100) [CPh$_3$]$^+$; elemental analysis calcd (%) for C$_{28}$H$_{21}$N$_2$O$_5$PW: C 49.44, H 3.11, N 4.12; found: C 49.22, H 3.20, N 4.11.

### Synthesis of complex 3d

A solution of a Li/Cl phosphinidenoid tungsten(0) complex **1** was prepared using 0.67 g (1.00 mmol, 1.0 eq.) of [W(CO)$_5${P(CPh$_3$)Cl$_2$}], 0.14 mL (0.87 mmol, 0.9 eq.) of 12-crown-4 and 0.77 mL ($c$ = 1.7 M in $n$-pentane, 1.31 mmol, 1.3 eq.) of a *tert*-butyllithium solution in 30 mL of THF at −80 °C. Afterwards, 0.17 mL (1.50 mmol, 1.5 eq.) of *tert*-butyl isocyanide (**2d**) was added dropwise at −50 °C. The reaction mixture was stirred for 16 h while it was allowed to slowly warm up to ambient temperature. All volatiles were removed in vacuo (<0.02 mbar) at ambient temperature. The product was extracted three times with 20 mL of diethyl ether. The solvent was removed in vacuo (<0.02 mbar) at ambient temperature and the product was obtained as yellow solid after drying under the same conditions for 2 h. Characterization data were consistent with literature values[62]. Yield: 0.56 g (0.82 mmol, 82%); yellow solid; $^1$H NMR (300.13 MHz, CDCl$_3$, 298 K): δ 7.33−7.24 (m, 15H), 1.28 (s, 9H); $^1$H NMR (500.14 MHz, C$_6$D$_6$, 298 K): δ 7.45−6.98 (m, 15H), 0.68 (s, 9H); $^{13}$C{$^1$H} NMR (125.76 MHz, CDCl$_3$, 298 K): δ 201.3 (d, $J$ = 16.9 Hz), 197.7 (d$_{sat}$, $J$ = 3.5 Hz, $J$ = 126.7 Hz), 146.8 (d, $J$ = 6.6 Hz), 143.0 (d, $J$ = 104.0 Hz), 130.4 (d, $J$ = 8.2 Hz), 128.1, 126.8 (d, $J$ = 1.1 Hz), 61.5, 58.4 (d, $J$ = 24.8 Hz), 29.7; $^{15}$N{$^1$H} NMR (50.69 MHz, C$_6$D$_6$, 298 K): δ −170.7; $^{31}$P (121.51 MHz, CDCl$_3$, 298 K): δ −50.5 (s$_{sat}$, $J$ = 117.5 Hz); IR (ATR diamond): 1889 (ν$_{CO}$), 1923 (ν$_{CO}$), 2060 (ν$_{CO}$), 2142 (ν$_{CN}$); MS (EI, 70 eV): $m/z$ (%) = 681 (0.1) [$M$]$^+$, 244 (100) [CPh$_3$ + H]$^+$.

### Synthesis of complex 10a

A solution of 0.233 g (0.34 mmol, 1.0 eq.) of complex **3c** and 0.38 mL (3.47 mmol, 10.1 eq.) of 1-pentene in 10 mL of benzene was stirred for 14 h at 70 °C. Afterwards, all volatiles were removed in vacuo (<0.02 mbar) at 70 °C. The obtained brown oil was further dried for 3 h at ambient temperature. The crude product was purified via column chromatography (SiO$_2$, ∅ = 1 cm, $h$ = 7 cm) using 200 mL of $n$-pentane. All volatiles were removed in vacuo (<0.02 mbar) at ambient temperature and the yellow oil was further dried for 1.5 h. Yield: 0.12 g (0.17 mmol, 50%); pale-yellow oil; isomeric ratio: 34:66 (*cis*: *trans*); $^1$H NMR (500.04 MHz, C$_6$D$_6$, 298 K): *cis*-**10a**: δ 7.45−7.42 (m, 6H), 7.10−6.98 (m, 9H), 1.71−1.63 (m, 1H), 1.60−1.54 (m, 2H), 1.06−0.88 (m, 3H), 0.58 (t, $J$ = 7.06 Hz, 3H), −0.12−−0.25 (m, 1H); *trans*-**10a**: δ 7.45−7.42 (m, 6H), 7.10−6.98 (m, 9H), 1.71−1.63 (m, 1H), 1.45−1.17 (m, 5H), 0.81 (t, $J$ = 7.14 Hz, 3H), 0.78−0.75 (m, 1H); $^{13}$C{$^1$H} NMR (125.75 MHz, C$_6$D$_6$, 298 K): *cis*-**10a**: δ 198.3 (d, $J$ = 32.9 Hz), 197.2 (d$_{sat}$, $J$ = 7.1 Hz, $J$ = 126.7 Hz), 142.6, 131.7 (d, $J$ = 7.7 Hz), 128.5, 127.7, 61.7 (d, $J$ = 3.2 Hz), 33.5 (d, $J$ = 18.8 Hz), 29.8 (d, $J$ = 3.7 Hz), 24.1 (d, $J$ = 6.7 Hz), 16.9 (d, $J$ = 18.3 Hz), 13.8; *trans*-**10a**: δ 197.4 (d, $J$ = 33.4 Hz), 196.8 (d$_{sat}$, $J$ = 7.2 Hz, $J$ = 126.6 Hz), 142.1, 131.5 (d, $J$ = 7.8 Hz), 128.3, 127.7, 62.1 (d, $J$ = 1.6 Hz), 34.1 (d, $J$ = 1.9 Hz), 23.0 (d, $J$ = 6.6 Hz), 22.9 (d, $J$ = 18.8 Hz), 17.3 (d, $J$ = 18.3 Hz), 13.8; $^{31}$P NMR (202.44 MHz, C$_6$D$_6$, 298 K): *cis*-**10a**: δ −137.7 (s$_{sat}$, $J$ = 258.8 Hz); *trans*-**10a**: δ −145.4 (d$_{sat}$, $J$ = 19.6 Hz, $J$ = 259.7 Hz); IR (ATR diamond): 1913 (ν$_{CO}$), 1983 (ν$_{CO}$), 2071 (ν$_{CO}$), 2959 (ν$_{CH}$); MS (LIFDI): $m/z$ (%) = 668.1 (100) [$M$]$^+$, 243.1 (10) [CPh$_3$]$^+$.

### Synthesis of complex 10b

A solution of 0.217 g (0.32 mmol, 1.0 eq.) of complex **3c** and 0.40 mL (3.20 mmol, 10.0 eq.) of 1-hexene in 12 mL of benzene was stirred for 16.5 h at 70 °C. Afterwards, all volatiles were removed in vacuo (<0.02 mbar) at ambient temperature. The crude product was purified via column chromatography (SiO$_2$, ∅ = 1 cm, $h$ = 10 cm) at ambient

temperature using *n*-pentane. Yield: 0.12 g (0.18 mmol, 55%); pale-yellow oil; isomeric ratio: 36:64 (*cis*: *trans*); $^1$H NMR (500.04 MHz, C$_6$D$_6$, 298 K): ***cis*−10b**: δ 7.45−7.43 (m, 6H), 7.10−7.05 (m, 6H), 7.03−6.97 (m, 3H), 1.74−1.65 (m, 1H), 1.62−1.53 (m, 2H), 1.28−1.18 (m, 2H), 1.16−1.09 (m, 1H), 1.03−0.92 (m, 2H), 0.70−0.67 (m, 3H), −0.09−−0.18 (m, 1H); ***trans*−10b**: δ 7.45−7.43 (m, 6H), 7.10−7.05 (m, 6H), 7.03−6.97 (m, 3H), 1.74−1.65 (m, 1H), 1.44−1.32 (m, 2H), 1.28−1.18 (m, 3H), 1.03−0.92 (m, 2H), 0.85−0.83 (m, 3H), 0.80−0.76 (m, 1H); $^{13}$C{$^1$H} NMR (125.75 MHz, C$_6$D$_6$, 298 K): ***cis*−10b**: δ 198.3 (d, $J$ = 32.9 Hz), 197.2 (d$_{sat}$, $J$ = 7.1 Hz, $J$ = 126.7 Hz), 142.6, 131.7 (d, $J$ = 7.7 Hz), 128.5, 127.7, 61.7 (d, $J$ = 3.2 Hz), 33.5 (d, $J$ = 18.8 Hz), 33.1 (d, $J$ = 6.6 Hz), 27.7 (d, $J$ = 3.8 Hz), 22.7, 17.1 (d, $J$ = 18.5 Hz), 14.0; ***trans*−10b**: δ 197.4 (d, $J$ = 33.3 Hz), 196.8 (d$_{sat}$, $J$ = 7.1 Hz, $J$ = 118.9 Hz), 142.2, 131.5 (d, $J$ = 7.9 Hz), 128.3, 127.7, 62.1 (d, $J$ = 1.5 Hz), 32.0 (d, $J$ = 6.5 Hz), 31.8 (d, $J$ = 1.9 Hz), 23.1 (d, $J$ = 18.8 Hz), 22.8, 17.4 (d, $J$ = 18.3 Hz), 14.1; $^{31}$P NMR (121.51 MHz, C$_6$D$_6$, 298 K): ***cis*−10b**: δ −137.4 (s$_{sat}$, $J$ = 258.9 Hz); ***trans*−10b**: δ −145.5 (d$_{sat}$, $J$ = 15.2 Hz, $J$ = 260.2 Hz); IR (ATR diamond): 1904 (ν$_{CO}$), 1982 (w) (ν$_{CO}$), 2070 (ν$_{CO}$), 2957 (ν$_{CH}$); MS (LIFDI): *m/z* (%) = 682.3 (100) [*M*]$^+$, 243.2 (5) [CPh$_3$]$^+$.

### Synthesis of complex 11a

0.63 mL (c = 2 M in THF, 1.26 mmol, 5.0 eq.) of a methylamine solution was added to a solution of 0.17 g (0.25 mmol, 1.0 eq.) of of complex **3d** in 20 mL of THF. The solution was stirred for 21 h at ambient temperature. Afterwards, all volatiles were removed in vacuo (<0.02 mbar) at ambient temperature. The product was isolated as pale-yellow solid by column chromatography (Al$_2$O$_3$, ∅ = 5 cm, *h* = 4 cm) using diethyl ether at ambient temperature. Yield: 0.03 g (0.05 mmol, 20%); pale-yellow solid; mp: 131−132 °C (dec.); $^1$H NMR (300.13 MHz, CDCl$_3$, 298 K): δ 7.36−7.11 (m, 15H), 6.78 (d, $J$ = 349.16 Hz, 1H), 3.22 (s, 1H), 2.96 (s, 3H), 1.04 (s, 9H); $^{13}$C{$^1$H} NMR (75.48 MHz, CDCl$_3$, 298 K): δ 198.1 (d, $J$ = 26.0 Hz), 196.3 (d, $J$ = 5.7 Hz), 151.7, 143.9, 129.6, 128.4, 127.5, 66.0, 58.9 (d, $J$ = 11.2 Hz), 29.9, 28.3; $^{15}$N{$^1$H} NMR (50.68 MHz, C$_6$D$_6$, 298 K): δ −144.6, −259.3; $^{31}$P NMR (121.51 MHz, CDCl$_3$, 298 K): δ −27.2 (d$_{sat}$, $J$ = 349.2 Hz, $J$ = 220.5 Hz); IR (ATR diamond): 1914 (ν$_{CO}$), 1988 (ν$_{CO}$), 2072 (ν$_{CO}$), 2372 (ν$_{PH}$), 3411 (ν$_{NH}$); MS (EI, 70 eV): *m/z* (%) = 712.0 (0.1) [*M*]$^+$, 684.0 (2) [*M* − CO]$^+$, 628.0 (0.1) [*M* − 3CO]$^+$, 599.9 (0.1) [*M* − 4CO]$^+$, 572.1 (0.1) [*M* − 5CO]$^+$, 542.9 (0.1) [*M* − 4CO-$^t$Bu]$^+$, 514.9 (0.1) [*M* − 5CO −$^t$Bu]$^+$, 440.9 (5) [*M* − CO − CPh$_3$]$^+$, 384.9 (2) [*M* − 3CO − CPh$_3$]$^+$, 356.8 (1) [*M* − 4CO − CPh$_3$]$^+$, 244.0 (100) [CPh$_3$ + H]$^+$, 166.0 (32) [CPh$_2$]$^+$, 113.0 (48) [C(NMe)N(H)$^t$Bu]$^+$, 77.0 (2) [Ph$^+$], 57.0 (65) [$^t$Bu]$^+$; elemental analysis calcd (%) for C$_{30}$H$_{29}$N$_2$O$_5$PW: C 50.58, H 4.10, N 3.93; found: C 52.26, H 4.60, N 3.60.

### Synthesis of complex 11b

0.251 g (0.37 mmol, 1.0 eq.) of complex **3d** was dissolved in 13 mL (159 mmol, 431 eq.) of isopropylamine at ambient temperature. The reaction mixture was stirred for 25 h at ambient temperature. All volatiles were removed in vacuo (<0.02 mbar) at ambient temperature. Yield: 0.19 g (0.26 mmol, 76%); pale-yellow solid; mp: 147−148 °C (dec.); $^1$H NMR (300.13 MHz, C$_6$D$_6$, 298 K): δ 7.59−6.99 (m, 15H), 6.64 (d, $J$ = 346.43 Hz, 1H), 3.33 (s, 1H), 3.27 (sept, $J$ = 6.09 Hz, 1H), 1.19 (s, 9H), 1.19 (d, $J$ = 6.09 Hz, 3H), 0.97 (d, $J$ = 6.09 Hz, 3H); $^{13}$C{$^1$H} NMR (125.78 MHz, C$_6$D$_6$, 298 K): δ 198.1 (d$_{sat}$, $J$ = 26.0 Hz, $J$ = 146.3 Hz), 196.6 (d$_{sat}$, $J$ = 5.7 Hz, $J$ = 126.3 Hz), 146.6, 144.0, 131.3−126.4 (m), 59.1 (d, $J$ = 10.2 Hz), 52.4 (d, $J$ = 6.3 Hz), 50.8 (d, $J$ = 11.4 Hz), 28.3, 24.8, 24.7; $^{15}$N{$^1$H} NMR (50.69 MHz, C$_6$D$_6$, 298 K): δ −112.4, −260.0; $^{31}$P NMR (121.51 MHz, C$_6$D$_6$, 298 K): δ −25.8 (d$_{sat}$, $J$ = 346.9 Hz, $J$ = 230.9 Hz); IR (ATR diamond): 1915 (ν$_{CO}$), 1987 (ν$_{CO}$), 2070 (ν$_{CO}$), 2379 (ν$_{PH}$), 3408 (ν$_{NH}$); MS (LIFDI): *m/z* (%) = 740.3 (100) [*M*]$^+$, 599.1 (3) [*M* − 5CO − H]$^+$, 417 (16) [*M* − W(CO)$_5$ + H]$^+$, 243.2 (82) [CPh$_3$]$^+$; elemental analysis calcd (%) for C$_{32}$H$_{33}$N$_2$O$_5$PW: C 51.91, H 4.49, N 3.78; found: C 52.22, H 4.53, N 3.59.

## Data availability

All data supporting the study can be obtained from the corresponding authors upon request. Experimental protocols as well as NMR spectra, cyclovoltammetric experiments and X-ray data results can be found in the Supplementary Information, including also computational methods and results. Coordinates of the optimized structures are present as source data. The X-ray crystallographic coordinates for structures reported in this study have been deposited at the Cambridge Crystallographic Data Centre (CCDC), under deposition numbers 2250849 (**3b**), 2250850 (**3$^{cr}$b**), 2250851 (**3c**), 2250852 (**3$^{cr}$c**), 2250853 (**3$^{cr}$d**), 2250854 (**11a**), 2250855 (**11$^{cr}$a**), 2250856 (**11b**) and 2250857 (**11$^{cr}$b**). These data can be obtained free of charge from the Cambridge Crystallographic Data Centre via www.ccdc.cam.ac.uk/data_request/cif. Source data are provided with this paper.

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

## Acknowledgements

We acknowledge financial support by the Deutsche Forschungsgemeinschaft (STR 411/45-1). A.E.F. wishes to acknowledge the computational resources used at the computation center at *Servicio de Cálculo Científico* (SCC – University of Murcia). R.T.B. thanks the University of Lethbridge for the award of Study Leave. This paper is dedicated to Prof. Alberto Tárraga on the occasion of his 70th birthday and his retirement.

## Author contributions

D.B. planned and carried out the experiments, collected and evaluated all analytical data. A.E.F. planned, conducted and analyzed all theoretical calculations. G.S. measured and evaluated the crystallographic data. R.S. had the supervision of this investigation. D.B., R.T.B., A.E.F. and R.S. wrote this paper. The SI was prepared by D.B., A.E.F. and R.S.

## Funding

## Competing interests

The authors declare no competing interests.
