## [Peer Review File · Nature Communications]

Challenging an old paradigm by demonstrating coordination and transition metal-like chemistry at a neutral nonmetal centerReviewers' Comments:

Reviewer #1:

Remarks to the Author:

It was found that low-coordinate heavier main-group species can behave like transition-metal complexes. Phosphinidenes have also been proven to display somewhat transition-metal-like behavior, such as oxidative addition, reductive elimination, and ligand substitution. This transition-metal like chemistry opens a new domain of main-group elements. In this work, Ferao and Streubel et al. reported the synthesis and ligand exchange reactions of the terminal phosphinidene complex as well as the addition of primary amines to the isocyanide phosphinidene complex. The latter reaction mimics the nucleophilic addition of a transition-metal complex (one of the elementary reactions of transition metals). The authors provide a deep insight into the process by employing NMR analysis, electrochemical experiments, and DFT calculations. Overall, this is a very fundamental and frontier work in low-coordinate phosphorus chemistry and will attract lots of interest from main-group chemists.

1. P1, Initially, a large range of complexes I possessing neutral..., but no label of complex I in Fig.1.
2. P8, the reversible coordination of phosphinidene complexes to alkenes and alkynes should be cited (Organometallics 2016, 35, 1170; Chem. Eur. J. 2019, 25, 15036).

Reviewer #2:

Remarks to the Author:

The authors present a series of neutral adducts (P-bound) of electrophilic tungsten phosphinidene complexes. The adducts are labile and in the case of isonitriles, have enhanced reactivity. These features are described by analogy to ligands bound to transition metal centers. The general topic of catalysis and reactivity of main group species is a current to great current interest. The work has been well executed and the results are presented in much detail. While there are some minor concerns about the presentation that can be readily addressed, the central question concerns the impact of the work - does it truly challenge an old paradigm?

1. Binding of ligands to metal-bound main group centers is not necessarily new (a).
2. An isocyanide ligand was activated towards additions of amines, which is somewhat novel.
3. The authors claim in final sentence of introduction "The detailed bonding analysis provides an understanding of key NMR features and computational mechanistic studies unveil parallels and differences between nonmetal and transition metal coordination chemistry for the first time." Upon reading this statement, one might have imagined data (exptl or theory) to directly compare reactivity, electrochemistry, and bonding of compounds 3d and 5d, so that impact of [(CO)5W] or [(CO)5W(RP)] binding to same isocyanide might be assessed. This report concentrates solely on [(CO)5W(RP)] adducts - the addition of comparative data could strengthen this communication.

(a) This paper places emphasis on isonitrile adducts at the metal-bound main group site (M-MG), and cite a number of studies where MG is silicon and the adduct is isocyanide. Although it is not made clear until later in the paper, the key compound, the isocyanide complex was previously reported by the authors (ref 62). Some of the text, however, leads one to believe it is new. For example, it is stated "The adducts 3b-d were stable at ambient temperature and could be all fully characterized, except pyridine adduct 3a, including single crystal X-ray diffraction studies of 3b-d (3c: Fig. 2; for other molecular structures from SC-XRD see SI-Fig. 111-115)."

Prior to other examples cited by Tokitoh, there are studies by Tilley showed reversible binding of acetonitrile at Si in ruthenium silylene complexes (Polyhedron, 1995, 148, 127). Gruner has also reported on cobalt carborane complexes where one of the metal bound boron atoms could bind nitriles, and that upon reaction with amines form products analogous to those reported in this submission (Dalton Trans., 2009, 851). Other examples have been reported, including some B-bound isonitrile and CO adducts of nickel carboranes (Inorganic Chemistry, 2005, 44, 8135).

(b) As one reads through paper, suddenly Cr complexes appear in section for electrochemistry. No discussion of their synthesis is given in main text, unlike for W complexes? Why?

(c) The authors state "All complexes 3a-d show significant ^{31}P NMR signatures (Table 1) which are related to some extent to those of Li/X phosphinidenoid complexes.⁷⁷ But the comparison also revealed that these belong to two significantly different families of adducts-to-phosphorus with respect to their ^{31}P NMR chemical shifts." It would be helpful to include the referenced data in Table so reader could understand what the "some extent" means.

(d) Scheme 3 presents some of the more exciting and novel results. P centered Lewis acids are well known, but formation of phosphirane complexes ("metal olefin complex" analogues) from the simple adducts is interesting and rare. Understanding and comparing "classic" substitution reactions and mechanisms would strengthen impact of manuscript.

Reviewer #3:

Remarks to the Author:

In this manuscript, Streubel and Ferao et al. investigated the synthesis and exchange reactions of labile ligand-to-phosphinidene complexes as well as the first transformation of a nonmetal-bound isocyanide to give either phosphaguanidine or carbene-to-phosphinidene complexes via 1,2 addition of primary amines to the P-C or the C-N bond, respectively. In addition, DFT calculations provide insights into the electronic structure as well as reaction mechanisms. These results might inspire and open new avenues in non-metal chemistry. However, the authors have to revise their manuscript to fully address the following issues before a final decision is reached.

1. The importance and the novelty should be highlighted not only in the abstract but also in the introduction, discussions and conclusions.
2. In the introduction, the authors claim that their computational mechanistic studies unveil parallels and differences between non-metal and transition metal coordination chemistry for the first time. But these were not discussed in the manuscript.
3. In page 12, the intermediates 14bE and 15b couldn't be found in corresponding figures, which should be mentioned in the manuscript even in the Supporting Information part.
4. The quality of Figure 8 is poor, which should be redrawn. Figure 9 could be moved to the supporting information.
5. In the theoretical part, many different theoretical methods were used, and the authors should explain the usage of different DFT method. In addition, it should be pointed out that the theoretical methods used in the wave function analysis. Were those analyses based on wave functions obtained from high-level single points?
6. There are some small mistakes in the manuscript, please double check the whole manuscript.
 - (1) In Figure 8, the energy barrier is 21.8 kcal/mol rather than 22.4 kcal/mol from 14cE to 15c.
 - (2) G should be italicized.
 - (3) The manuscript uses two kind of units for energy, kcal/mol and kJ/mol, which should be unified.

Response to Reviewers Comments

Reviewer #1 (Remarks to the Author):

It was found that low-coordinate heavier main-group species can behave like transition-metal complexes. Phosphinidenes have also been proven to display somewhat transition-metal-like behavior, such as oxidative addition, reductive elimination, and ligand substitution. This transition-metal like chemistry opens a new domain of main-group elements. In this work, Ferao and Streubel et al. reported the synthesis and ligand exchange reactions of the terminal phosphinidene complex as well as the addition of primary amines to the isocyanide phosphinidene complex. The latter reaction mimics the nucleophilic addition of a transition-metal complex (one of the elementary reactions of transition metals). The authors provide a deep insight into the process by employing NMR analysis, electrochemical experiments, and DFT calculations. Overall, this is a very fundamental and frontier work in low-coordinate phosphorus chemistry and will attract lots of interest from main-group chemists.

1. P1, Initially, a large range of complexes I possessing neutral..., but no label of complex I in Fig.1.

Answer: The Roman numeral has been removed from the text to avoid misunderstandings.

2. P8, the reversible coordination of phosphinidene complexes to alkenes and alkynes should be cited (Organometallics 2016, 35, 1170; Chem. Eur. J. 2019, 25, 15036).

Answer: The literature of Slootweg & Lammertsma, and Tian & Duan have been added to the discussion:

“At this point it should also be noted that e.g. π -systems such as alkenes can coordinate reversibly to phosphinidene complexes to form respective phosphirane complexes if terminal alkenes are used as solvent.^{86,87”}

Reviewer #2 (Remarks to the Author):

The authors present a series of neutral adducts (P-bound) of electrophilic tungsten phosphinidene complexes. The adducts are labile and in the case of isonitriles, have enhanced reactivity. These features are described by analogy to ligands bound to transition metal centers. The general topic of catalysis and reactivity of main group species is a current to great current interest. The work has been well executed and the results are presented in much detail. While there are some minor concerns about the presentation that can be readily addressed, the central question concerns the impact of the work - does it truly challenge an old paradigm?

1. Binding of ligands to metal-bound main group centers is not necessarily new (a).

Answer: We have added some more examples of isolated adducts from main group metalloid elements such as Silicon and Boron to provide a broader picture.

2. An isocyanide ligand was activated towards additions of amines, which is somewhat novel.

Answer: This reaction is one of the early textbook examples in Organometallic chemistry describing the conversion of an isocyanide into a carbene ligand. But it is novel, as the reviewer underlines, when attached to a nonmetal center.

3. The authors claim in final sentence of introduction "The detailed bonding analysis provides an understanding of key NMR features and computational mechanistic studies unveil parallels and differences between nonmetal and transition metal coordination chemistry for the first time." Upon reading this statement, one might have imagined data (exptl or theory) to directly compare reactivity, electrochemistry, and bonding of compounds **3d** and **5d**, so that impact of [(CO)₅W] or [(CO)₅W(RP)] binding to same isocyanide might be assessed. This report concentrates solely on [(CO)₅W(RP)] adducts - the addition of comparative data could strengthen this communication.

Answer: We have added the examples of *cis*-[PdCl₂(CNAr)(PPh₃)] (Ar = Ph, *p*-tolyl, *p*-C₆H₄Cl), [Fe(CNMe)₅] and [Ru(CNMe)₆] (ref. 22,24) additionally in a brief comment, which were also mentioned in the introduction.

(a) This paper places emphasis on isonitrile adducts at the metal-bound main group site (M-MG), and cite a number of studies where MG is silicon and the adduct is isocyanide. Although it is not made clear until later in the paper, the key compound, the isocyanide complex was previously reported by the authors (ref 62). Some of the text, however, leads one to believe it is new. For example, it is stated "The adducts **3b-d** were stable at ambient temperature and could be all fully characterized, except pyridine adduct **3a**, including single crystal X-ray diffraction studies of **3b-d** (3c: Fig. 2; for other molecular structures from SC-XRD see SI-Fig. 111–115)."

Answer: In the beginning of the discussion (p. 4), when the isocyanide adduct **3d** appears first, an additional sentence has been added for clarification and to avoid any misunderstanding. This clearly states that compound **3d** was already obtained earlier but using a different pathway, and less efficiently:

"The isocyanide adduct **3d** was obtained previously in lower yields from 3-iminoazaphosphiridine complexes via an exchange reaction.⁶⁸

Prior to other examples cited by Tokitoh, there are studies by Tilley showed reversible binding of acetonitrile at Si in ruthenium silylene complexes (Polyhedron, 1995, 148, 127).

Answer: The report of Tilley and coworkers on a nitrile-to-silylene complex adduct has been included in the introduction:

"[...] metalloid-bound nitrile adducts have been reported for silylenes [...] by Tilley for a cationic silylene ruthenium(II) complex²⁵ and Tokitoh for a neutral unligated silylene.²⁶"

Gruner has also reported on cobalt carborane complexes where one of the metal bound boron atoms could bind nitriles, and that upon reaction with amines form products analogous to those reported in this submission (Dalton Trans., 2009, 851). Other examples have been reported, including some B-bound isonitrile and CO adducts of nickel carboranes (Inorganic Chemistry, 2005, 44, 8135).

Answer: An additional statement for boron adducts was made in the introduction including the literature of Stone and Gruner mentioned by Reviewer #2 as well as further reports of the groups of Stephan, Bertrand and Braunschweig. However, also mentioning group 13 element compounds such as boranes as special case due to its electronic structure, and the reaction of a carborane nitrile adduct with amines was also included in the revised manuscript:

"As far as we know, adducts of low-coordinate boron centers as examples of group 13 metalloids are extremely rare.^{40–44} But classical boranes, having all valence electrons involved in covalent bonding, can

also form Lewis acid adducts with nitriles, isocyanides and carbon monoxide due to a vacant p-orbital. Of special interest is one example from B,C-cluster chemistry, i.e., nitrile-to-carborane complex adducts react with secondary amines in a nucleophilic addition to form acetamidine derivatives.⁴¹”

Comment: While the first step, the attack of the amine at the carbon atom, is similar to our reported case, neither a proton shift to B or N was observed nor the final compounds are same, i.e., (phospha)guanidines or carbene adducts are not accessible via this pathway.

(b) As one reads through paper, suddenly Cr complexes appear in section for electrochemistry. No discussion of their synthesis is given in main text, unlike for W complexes? Why?

Answer: An additional statement describing that the complex was obtained analogously was added after the discussion of the synthesis of tungsten complexes **3b-d**.

“Respective chromium complexes can be obtained analogously (see ESI).”

Comment: The adducts were also isolated as chromium complexes **3b-d**^{Cr} and characterized, as well as the phosphaguanidine complexes **11a,b**^{Cr}. However, neither a different reactivity could be observed nor any additional information could be obtained from the chromium case. Since the tungsten complexes revealed additional information in the NMR spectra via *J* coupling, the authors focused on the W derivatives in the main text – except for the electrochemical study to disclose the observed trends in a more reliable manner. Nevertheless, the synthesis and characterization of the Cr complexes can be found in the ESI.

(c) The authors state "All complexes 3a-d show significant ³¹P NMR signatures (Table 1) which are related to some extent to those of Li/X phosphinidenoid complexes.⁷⁷ But the comparison also revealed that these belong to two significantly different families of adducts-to-phosphorus with respect to their ³¹P NMR chemical shifts." It would be helpful to include the referenced data in Table so reader could understand what the "some extent" means.

Answer: The table was revised accordingly and the ³¹P NMR data of the Li/Cl phosphinidenoid complex **1** were included.

(d) Scheme 3 presents some of the more exciting and novel results. P centered Lewis acids are well known, but formation of phosphirane complexes ("metal olefin complex" analogues) from the simple adducts is interesting and rare. Understanding and comparing "classic" substitution reactions and mechanisms would strengthen impact of manuscript.

Answer: We agree with the reviewer and a complementary computational study on two model reactions were included in more detail (they were previously cited in the last paragraph, before Conclusions) now at the beginning of the Theoretical results, as follows.

„For a better understanding of the reactivity of the electrophilic phosphorus centre both in ligand substitution reactions at the P atom and in the formation of complexed phosphiranes, quantum chemical calculations at the COSMO_{THF}/CCSD(T)/def2-TZVPP(ecp) level (see the SI for computational details^{91,92}) were performed for two model reactions of the most labile *N*-methylimidazole adduct and using the simplified *P-tert*-butyl substituted pentacarbonyltungsten(0) complex model **3c**” for the sake of computational efficiency (Fig. 7). Ligand exchange reaction at P with (model) MeNC ligand takes place exergonically by S_N2 associative mechanism over a rather low barrier. However, reaction of **3c**” with ethylene as model olefin requires endergonic barrierless dissociation of the *N*-Melm ligand,

forming model phosphinidene **4''** as true intermediate, followed by barrierless chelotropic cycloaddition affording the model phosphirane complex **10c''** in an overall exergonic process (Fig. 7). For the latter step, a low barrier was found previously in case of P-amino-substituted phosphinidene complexes.⁹³ Complex **14** could also be formed by direct dipolar cycloaddition reaction of **3c''** with ethylene, although this is unfavourable due to its slightly lower stability and the higher energy barrier of this step. Indeed, such bicyclic derivative has never been observed. Interestingly, almost identical values were obtained at the much faster PBEh-3c optimization level (see the SI)."

Fig. 7 Proposed mechanism for the reactions of model *N*-Melm adduct complex **3c''** with methyl isocyanide and ethylene. Computed [CPCM_{toluene}/CCSD(T)/def2-TZVPP(ecp)//CPCM_{toluene}/B3LYP-D3/def2-TZVP(ecp)] relative Gibbs free energies (kcal/mol) in red and square brackets.

Reviewer #3 (Remarks to the Author):

In this manuscript, Streubel and Ferao et al. investigated the synthesis and exchange reactions of labile ligand-to-phosphinidene complexes as well as the first transformation of a nonmetal-bound isocyanide to give either phosphaguanidine or carbene-to-phosphinidene complexes via 1,2 addition of primary amines to the P-C or the C-N bond, respectively. In addition, DFT calculations provide insights into the electronic structure as well as reaction mechanisms. These results might inspire and open new avenues in non-metal chemistry. However, the authors have to revise their manuscript to fully address the following issues before a final decision is reached.

1. The importance and the novelty should be highlighted not only in the abstract but also in the introduction, discussions and conclusions.

Answer: The Abstract has been completely re-written and the novelty of the work has been emphasized in the Introduction, discussion and conclusions. We are grateful to the Reviewer for recognizing the importance of our results and conclusions.

2. In the introduction, the authors claim that their computational mechanistic studies unveil parallels and differences between non-metal and transition metal coordination chemistry for the first time. But these were not discussed in the manuscript.

Answer: We have added the following text after the discussion of the theoretical results on the reactions of **3d** with primary amines:

"It should be noted here that the calculations not only support the experimental results but also clearly reveal differences between the chemistry of isocyanide metal complexes and the P-adducts reported here as the former do not exhibit any 1,2-addition reactivity to the M-C(ligand) bond while in case of

the P-C(ligand) bond it is the preferred pathway for small primary amines. Only if sterics become important the organometallic-like reactivity of the phosphorus center comes to the fore, i.e., the isocyanide-to-carbene ligand conversion via 1,2-addition to the C-N bond.”

Additionally, we added the following to the conclusions part to underline our initial statement:

“The case of primary amines also clearly reveal differences to metal-bound ligand conversions as a 1,2-addition of the N-H bond to an M-C bond of an isocyanide ligand is not preferred while in case of phosphorus it is for less encumbered amines. If sterics become more important than the parallel in reactivity comes to the fore, the 1,2-addition to the C-N bond.”

3. In page 12, the intermediates 14bE and 15b couldn't be found in corresponding figures, which should be mentioned in the manuscript even in the Supporting Information part.

Answer: Derivatives named as **Xb** refers to those derived from the reaction with isopropylamine and therefore no comment is needed in Figure 7 which is valid for the reactions of all three amines. On the contrary Figure 8 (energy profile) collects data for the two limiting cases of sterically non-demanding (methyl, “a”) and demanding (*tert*-butyl, “c”) amines. Isopropylamine (“b”) very much behaves as methylamine and thus was not included in Figure 8 (to keep it as simple as possible) and only significant variations were commented in the text. The structures and energies for isopropylamine, **15b^F**, **16b** and **16b^P** (updated numbering) were collected in the submitted Supporting Information file. To make it clear, we have added the statement “(not shown)” when referring to these compounds in the main text.

4. The quality of Figure 8 is poor, which should be redrawn. Figure 9 could be moved to the supporting information.

Answer: The resolution of Figure 8 has been improved, and Figure 9 has been moved to the SI, as per the Reviewer's advice.

5. In the theoretical part, many different theoretical methods were used, and the authors should explain the usage of different DFT method. In addition, it should be pointed out that the theoretical methods used in the wave function analysis. Were those analyses based on wave functions obtained from high-level single points?

Answer: We thank the Reviewer for raising this important issue. First of all, all structures were optimized at the B3LYP-D3/def2-TZVP using solvent continuum models for all mechanistic studies. Although the standard method used for solvent effects was CPCM, for the mechanistic study of the reaction of the isocyanide complex adduct with amines, COSMO(THF) was employed, in analogy with previous reports from us (e.g. *Chem. Commun.*, **2020**, 56, 3899; *Dalton Trans.*, **2021**, 50, 739). We checked selected reactions using the CPCM(THF) alternative and found negligible energy differences.

Regarding final energy calculations, most of them were computed at the “gold standard” CCSD(T)/def2-TZVPP level, except for the more demanding trityl-substituted (real) derivatives, for which the computationally less costly double hybrid PWPB95-D3 functional was used, together with the def2-QZVPP(ecp) basis set. The latter was also used for TOP and FIA calculations in order to allow the calculation of trityl-substituted derivatives.

Chemical shift values were performed at the GIAO/PBE0/def2-TZVP(ecp) level in order to use previously reported well correlated conversion procedures with a wide set of experimental data (e.g. *Inorg. Chem.* **2021**, *60*, 13029)

NBO and AIM wavefunction analyses were performed at the B3LYP/def2-TZVPP level with NBO 6.0 and Multiwfn 3.7, respectively. This information was missing in our first submission and has been added in the SI.

HOMO/LUMO energies and Mulliken charges were obtained at the PWPB95-D3/def2-QZVPP(ecp) level. These distinctions have been clearly spelt out in the Computational Details section within the Theoretical Investigations report that forms a major part of the Supporting Information, and appropriate Call Outs to this are found in the main article (e.g. on p.12).

6. There are some small mistakes in the manuscript, please double check the whole manuscript.

Answer: The whole manuscript has been thoroughly edited for clarity and correctness. All substantive changes, and direct responses to reviewer comments, are highlighted in the re-submitted version.

(1) In Figure 8, the energy barrier is 21.8 kcal/mol rather than 22.4 kcal/mol from 14cE to 15c.

Answer: The reviewer is right, and the value was corrected in the text.

(2) G should be italicized.

Answer: Variables such as *G* were revised accordingly.

(3) The manuscript uses two kind of units for energy, kcal/mol and kJ/mol, which should be unified.

Answer: The whole structure of the manuscript and all the associated figures embed kcal/mol because of its widespread use in computational chemistry. But for the FIA and TOP calculations, we presented the outcomes in kJ/mol units to facilitate comparison with the standard reference publications on these topics. To appease the concern of Reviewer 3, we have provided the energies for our key FIA and TOP results in kcal/mol alongside the original values.

Reviewers' Comments:

Reviewer #2:

Remarks to the Author:

The authors have done a thorough job revising the manuscript in light of the reviewers comments, and in doing so, significantly improved the manuscript. It can now be recommended for publication.

Reviewer #3:

Remarks to the Author:

The authors have revised the manuscript based on my comments, this manuscript could be acceptable now.